# Genetic Approaches for the Treatment of Giant Axonal Neuropathy

**DOI:** 10.3390/jpm13010091

**Published:** 2022-12-30

**Authors:** Satomi Shirakaki, Rohini Roy Roshmi, Toshifumi Yokota

**Affiliations:** Department of Medical Genetics, University of Alberta, Edmonton, AB T6G 2H7, Canada

**Keywords:** GAN, gigaxonin, ubiquitin-proteosome system, axon, gene therapy, AAV9, autophagy

## Abstract

Giant axonal neuropathy (GAN) is a pediatric, hereditary, neurodegenerative disorder that affects both the central and peripheral nervous systems. It is caused by mutations in the *GAN* gene, which codes for the gigaxonin protein. Gigaxonin plays a role in intermediate filament (IF) turnover hence loss of function of this protein leads to IF aggregates in various types of cells. These aggregates can lead to abnormal cellular function that manifests as a diverse set of symptoms in persons with GAN including nerve degeneration, cognitive issues, skin diseases, vision loss, and muscle weakness. GAN has no cure at this time. Currently, an adeno-associated virus (AAV) 9-mediated gene replacement therapy is being tested in a phase I clinical trial for the treatment of GAN. This review paper aims to provide an overview of giant axonal neuropathy and the current efforts at developing a treatment for this devastating disease.

## 1. Introduction

Giant axonal neuropathy (GAN, OMIM #256850) is a rare neuromuscular disease with an autosomal recessive mode of inheritance [1]. The disease presents as a prominent sensorimotor neuropathy in early childhood and commonly progresses to affect both the peripheral and central nervous systems (PNS and CNS) [2,3]. Most children become wheelchair-dependent in the second decade of life and typically die in the third decade from respiratory failure [4,5,6].

GAN is caused by a mutation in the GAN gene, which encodes a protein called gigaxonin [1]. Gigaxonin is a part of the ubiquitin-proteasome system and controls autophagosome production [7]. Specifically, gigaxonin plays an important role in the breakdown of neurofilament (NF) [8], where the cellular hallmark of GAN pathology is the formation of its large aggregates [2,3,5]. According to the Leiden Open Variation Database (LOVD) (updated 2022), National Center for Biotechnology Information (NCBI) (updated 2022), and the mutation summary reported by Lescouzères and Bomont, P. (2020), there are 89 variants that lead to manifestation of GAN (Figure 1) [5,8,9,10,11,12,13,14,15,16,17,18,19,20,21,22,23,24,25,26,27,28,29,30,31,32,33,34,35,36].

Presently there is no effective treatment for GAN. Only symptomatic management to slow the progression of the disease are in place [4]. A combination of speech, occupation, and physical therapy is used by physicians to enhance physical and cognitive development. As such, there is a need for a therapy that provides an effective yet safe treatment that addresses the primary cause of GAN.

Here, we summarize current understanding and genetic approaches that have been explored for the treatment of GAN. In addition, we will discuss the difficulties they may be facing on their way from preclinical studies to clinical translation and offer some future perspectives.

## 2. Clinical Heterogeneity in Persons with GAN

Since the first documented case in 1972, several dozens of patients with GAN have been reported, with a strong clinical heterogeneity and diverse prognosis [2,3,22]. The classic GAN typically manifests as an infantile to early-childhood onset neurodegenerative disorder with a decline in motor and sensory function. During infancy, clinical assessment usually reveals motor and sensory neuropathy with moderate axonal degeneration (Table 1) [2,3,22,23]. Progressive distal motor weakness is the initial symptom in all cases. Diffused muscle atrophy, most predominantly in distal muscles, flaccid, paralysis, severely decreased muscle strength, low muscle tone, and loss of reflexes (areflexia) may be observed as the disease advances (Table 2) [3,22,23,24,25,26,37,38]. Diagnosis of PNS degeneration in early infancy and sensorimotor pathway involvement in teens resemble more commonly inherited peripheral neuropathy called Charcot-Marie-Tooth (CMT) diseases [27]. Owing to similarities in clinical presentation between patients with GAN and CMT or Friedreich ataxia, CTM Disease Pediatric Scale and Friedreich Ataxia Rating Scale and Gross Motor Function measure may be implemented for physical examination [4,37]. Though, making PNS deterioration unique from some of the other peripheral neuropathies, GAN also leads to proximal motor weakness, evidenced by pectoralis chest-wall muscle wasting, winged scapula, and exhibition of myopathic or “waddling” gait disturbances and positive Gower’s sign during the attempt to achieve erect position, indicative of pelvic girdle and quadriceps muscle weakness [2,4,6,25,28]. Consistent with PNS clinical presentation, an electromyogram (EMG) demonstrates neuropathic changes indicative of multiple peripheral nerve damage with chronic denervation, mainly involving the sensorimotor branches of limbs [29]. Nerve conduction studies show markedly decreased to absent compound muscle action potentials and sensory nerve action potentials of the upper and lower limb, and prolonged motor and sensory nerve conduction velocities, even to the demyelinating range [23,25,26,29,39]. The electron microscopy of the sural nerve and skin/muscle fiber biopsy shows abnormally large axons in their paranodal junction and decreased myelinated axons (Table 1) [23,26,40] The general morphology of nerve fibers may be maintained, although lacks normally identifiable sidearms that extend from healthy NF from increased NF packing, which is critical for modulating NF spacing and managing the mechanical integrity of nerve cells [4,41]. The enlarged axons exhibit significantly decreased myelin sheath thickness and are filled with disorganized NF aggregates, distending axon and axoplasm, and widening periaxonal space [26,41]. This causes the pathological hallmark of GAN, a ‘giant’ axon. These axons also contain a reduced number of microtubules and other remaining axonal organelles such as mitochondria are pushed out into sub-axolemmal space [4,6,41]. The swelling can occur both in myelinated and unmyelinated axon fibers, often beginning at the node of Ranvier and having a segmental swelling appearance [2,4]. Some fibers can be found surrounded by basal lamina, which has an important role in regeneration and remyelination [22,42]. This agrees with EMG that shows partial reinnervation beside demyelinated or lesions fibers, especially in the patient with CMT-like presentation [22,29]. Not limited to the disorganization of NF, other types of disorganized intermediate filament (IF) accumulation can be found in various peripheral nerves and cells.

GAN also differs from other neuropathies due to CNS involvement including pyramidal signs, positive Babinski signs indicative of deficit in the upper motor neuron, and positive Romberg signs associated with dorsal column lesion (Table 2) [2,22]. Among these, cerebellar involvement is most extensive, including truncal ataxia, incoordination, fine movement impairment, tremor, dysmetria, nystagmus, and oculomotor apraxia [25,37,39]. Cranial nerve impairment is also described in patients with GAN, causing facial weakness, ptosis, and ophthalmoplegia, involving facial, oculomotor, trochlear, and abducens nerves [6,26,29,30]. Many patients have vision loss with optic atrophy, as well as dysarthria, dysphonia, dysphasia, and hearing impairment, reflecting the combination of both central and peripheral dysfunction [25,30]. Severe CNS involvement may include vertigo, intellectual disability, spasticity, seizures/epilepsy, and dementia (Table 2) [4,29,39,40]. Neuroimaging study demonstrates slowly progressive diffuse leukoencephalopathy with demyelination and atrophy, and signal abnormalities throughout cerebrum, cerebellum, and notably in spinal cord [22,30,41]. One of the early signs in disease is the signal abnormalities (increased T2 signal intensity) detected with magnetic resonance imaging (MRI) [25,26,39]. T2 hyperintense lesion involve internal capsule, globus pallidus of the basal ganglia, thalami, brainstem. and spinal cord, where frontoparietal and periventricular white matter and dentate nucleus of cerebellum affected most prominently [25,43]. T1 hyperintense lesion and decreased T1 signal at varying parts of the brain associated with brain atrophy may be also observed. Similarly, the fractional anisotropy value is low reflecting increased diffusivity due to intracellular water increase from axonal distension and demyelination [43]. Spinal cord atrophy in posterior column, medial lemniscus spinocerebellar tracts, and cortical spinal tracts, involving abnormality in inferior olivary, gracile and cuneate nuclei, and cerebellar peduncles along pathways are prominent [40,43]. As axonal transport is impaired, peripheral nerves, posterior columns, cerebellum and pyramidal tracts are most severely affected [4]. Besides axonal loss, loss of Purkinje cells and other neuronal cells are also reported [40]. Secondary demyelination may also be present due to nerve cell body loss [4,26]. In addition to IF aggregations in CNS that leads to aforementioned axonal demyelination and atrophy, there is also a prominent presence of Rosenthal fibers which is an astrocyte pathology described in Alexander disease [6,23,40]. The electroencephalogram (EEG) study further supports clinical signs. Evoked potentials show increased latency in auditory, visual, and somatosensory evoked response, suggesting lesion in brainstem and higher cortices [44]. EEG may also show epileptiform transients discharges in the form of spikes and sharp waves in patient with or without history of epilepsy (Table 1) [26,29]. GAN also alters brain metabolites. Major detectable brain metabolites with magnetic resonance spectroscopy (MRS) include the predominantly neuroaxonal compound *N*-acetylaspartate (NAA), the energy metabolites creatine and Cho compounds involved in membrane turnover, and osmolyte myoinositol (Ins) [43]. The Cho and Ins typically increase with respect to demyelination and glial proliferation while NAA decreases with axonal damage/loss [43]. While these typical GAN clinical findings are prevalent, individuals with genetically confirmed GAN can be affected more mildly [4,22]. Milder cases can be regarded as CMT-plus phenotype and is becoming increasingly recognized. The disease may present with slow onset and progression of peripheral neuropathy with minimal CNS involvement. [29,39,40,41].

GAN is also associated with gastrointestinal (GI) and systemic issues including constipation, reflux, regurgitation, diabetes, renal tubular acidosis, and lactose intolerance [30,32,34,45,46].Some patients may also suffer from precocious puberty, arched feet, scoliosis, tendon contracture, and lumbar hyper-lordosis [8,22,23,25,29]. GAN patients often have a characteristic physical appearance with kinky hair, long eyelashes, a high forehead, pale skin, and facial diplegia [26]. Kinky hair and long eyelashes are frequently observed features but not in all cases and are most likely due to abnormal keratin accumulation (Table 2) [32]. Ichthyosis and keratosis pilaris can also occur, although minor [21]. Despite heterogenicity in phenotype within GAN, the relationship between phenotypic differences and the rate of disease progression is yet to be known. 

## 3. Gigaxonin Plays an Important Role in IF Homeostasis

Disorganization of the NF network is a key feature of several neurodegenerative disorders. Likewise, disorganized NFs is prominent in GAN. Though, GAN is unique in that GAN corresponds to a generalized disorganization of the all classes of cytoskeletal IFs [47]. Dysregulated IFs in GAN axons include keratin (class I and II IFs), vimentin, desmin, glial fibrillary acidic protein (GFAP) and peripherin (class III), and NF (class IV) [35,48]. Besides the NF accumulation in axons, aggregates can be seen within peripheral sensorimotor nerves, as well as Schwan cells, endothelial cells, perineurial cells, lens epithelial cells, skin fibroblasts, and muscle fibers [47]. Accumulation also occurs in CNS involving cerebral and cerebellar white matter, middle cerebellar peduncles, brainstem tegmentum, corticospinal tracts, and posterior column, as IFs accumulate in astrocytes, neurons of white matter and spinal tract, supporting white matter abnormalities reported in imaging [39,43]. In support of autonomic nervous system involvement seen with symptoms including vomiting and constipation, IF can also aggregate in the myenteric plexus of the GI tract [39,45,46]. In particular, NFs plays important role in radial growth for achieving rigid cytoskeletal structure and stability of myelinated axons to allow optimal electrical impulse propagation along axons [49]. NF consists of aforementioned sidearms in the form of subunits (NF—light (L), medium (M), and heavy (H) chain, and a-internexin) that allows cross-link to other cellular organelles [49]. Reduction in NFs inhibits axonal radial growth or cross-bridge and interconnection with other non-IF components of the cytoskeleton including microtubules and actin-filament and disrupts synaptic plasticity [49].

In GAN, IF disorganization results from mutations in the *GAN* gene. The *GAN* gene encompasses 11 exons located in chromosome 16 and encodes a 65-kDa protein called gigaxonin, which is normally present in cells throughout the body at a low level and strongly expressed in the brain, heart, and skeletal muscle [36]. At least 150 different mutations in the gene have been described. [3], and the disease-causing mutations that lead to the loss of functional gigaxonin or production of abnormal gigaxonin can span the entire coding region of the *GAN* gene [7,36]. Mutations can reduce mRNA quantity decreasing gigaxonin abundance and destabilizing protein structure by improper folding, impairing its activity and reducing half-lives [35].

Gigaxonin is comprised of an *N*-terminal BTB (Broad-Complex, Tramtrack, and Bric a brac) domain, a C-terminal Kelch repeat domain, and a BACK domain between the terminus ends and is considered to belong to the BTB/kelch protein family [36,50]. Structural and functional studies demonstrate that they are a substrate-specific adaptor protein for a Cul3-dependent/gigaxonin E3 ubiquitin ligase complex [36,50,51]. Combined with sequence homology, gigaxonin appears to be involved in protein degradation via ubiquitin proteasomal-dependant pathway, implicated in the cytoskeletal network [48,51]. Relevant in the GAN context, a mutation in BTB-Kelch gigaxonin protein leads to IF accumulation. Major subunits of the ligase complex are cullin, RING box protein (Rbx), and an adaptor that directs the substrate near the E2 enzyme, recruited by Rbx [52]. BTB domain of gigaxonin interacts with Rbx1 and directly binds with cullin 3 (Cul3), both a component of E3 ubiquitin ligase complexes, to associate the two to form functional complex [52,53,54,55]. Direct interaction between the BTB domain and Cul3 suggests the interaction between proteasome components [55]. The Kelch domain recruits substrates for polyubiquitination, mediating the addition of ubiquitin chains onto their targets, before their degradation by the proteasome [47,49]. Ubiquitination is a post-translational, enzymatic modification process where ubiquitin-protein binds substrate protein to destine cell fate and modulate cellular function and signaling pathway [7,56]. Subsequently, polyubiquitination, formation of ubiquitin chain, initiates proteolysis and degradation of protein to maintain homeostasis [56]. In particular, gigaxonin control microtubule dynamics as Kelch targets microtubule-associated proteins (MAP), including microtubule-associated protein 1B (MAP1B), tubulin-folding cofactor B (TBCB), and microtubule-associated protein 8 (MAP8), as its binding partner [4]. MAP is involved in many cellular processes including morphogenesis and differentiation, to maintain cytoskeletal integrity [57]. Gigaxonin physically colocalizes with MAP in neurons along the microtubule network and binds directly to MAPs for it to be targeted by the ubiquitin-proteasome system for its degradation. [51,58,59,60]. This interaction is important in the enhancement of microtubule stability, as destabilized microtubules interfere with cytoskeletal balances and aggravate IF aggregation which impairs axonal transport, affecting neuronal function and survival [51,59]. Individually, MAPs can cause neurodegeneration and death when overexpressed, and the survival rate improves when silenced in GAN neurons, as well. Though, functions of gigaxonin focused on MAP have not yet provide a causal link between the loss of functional gigaxonin and IF aggregate formation in patients. For instance, vimentin IF aggregates are not produced upon microtubule clearance in control fibroblasts [7]. Moreover, TBCB that interacts with certain mutated gigaxonin can retain its level of activity [7]. Despite causal microtubule implication in the pathogenesis of GAN cannot be established, gigaxonin plays an important role in controlling microtube dynamics such that *GAN* mutations impair IF degradation, resulting in an excess IFs in axons and promotes aggregation [35]. In addition, restoration of gigaxonin induces degradation of several IFs [56,61].

Recent evidence also shows gigaxonin-E3 ligase involvement in regulating the autophagy pathway and autophagosome production [7,62]. A third of E3 ligase is involved in the autophagy pathway and mediates ubiquitin-dependent protein destruction by autophagy. Autophagy is a process that delivers cytoplasmic material, such as damaged organelles, ribosomes, and protein aggregates, to the lysosome for degradation [62]. This degradation pathway is important in cell survival and the maintenance of bioenergetic homeostasis. Materials to be degraded need to be surrounded by a phagophore (incomplete autophagosome) which elongates to form a complete autophagosome that can bind with a lysosome. ATG16L1 plays a role in specifying the LC3 lipidation site and thus, is a key determinant for the elongation of the phagophore. As per IFs aggregate, large perinuclear bundles of ATG16L1 can form in some of the neurons under gigaxonin depletion conditions. Alternation in ATG16L1 activities or accumulated ATG16L1 inhibits autophagosome formation by impairing its maturation. This leads to phagophore accumulation and p62 aggregation, which is the main autophagy receptor, due to defective autophagic degradation and diminished autophagosome-lysosome events, impairing autophagic flux. Precisely, GAN neurons can generate autophagosomes but lack the capability of maintaining the production over prolonged induction of autophagy [7,62]. Several studies suggest a role of gigaxonin also in cell metabolism, where nutrient-responsive gigaxonin glycosylation forms the regulatory link between metabolism and IF turnover [63].

## 4. In Vivo Gene Therapy for GAN Using AAVs

As loss of function mutations in the *GAN* gene result in GAN, gene replacement therapy to express functional gigaxonin is a potential option for therapeutics [64]. It is expected that the expression of functional gigaxonin will reduce the pathogenic IF aggregates in patients. Gray et al. have established that treatment with a normal copy of human GAN transgene (AAV9/JetT-GAN) restores the normal pattern of IF distribution in fibroblasts obtained from a person with GAN, as well as significantly reduces aggregates of vimentin within 3 days of treatment [64]. Consistent with their in vitro findings, aged GAN knockout (KO) mice exhibited reduced NF aggregates within 4 weeks of treatment with the *GAN* replacement vector. Additionally, intrathecal (IT) injection of the *GAN* replacement vector sustained gigaxonin expression in the CNS and PNS of GAN KO mice for up to a year and exhibited conservation of sciatic nerve architecture. Treatment also showed improved rotarod performance in GAN KO mice. However, the improvement in rotarod function was not seen in mice aged 23 months. The authors speculate that the treatment dose may not have been optimal to fully rescue rotarod function. No toxicity from treatment with AAV9/JetT-GAN is reported in vivo.

In this study, the authors also address the concern of causing toxicity due to the overexpression of gigaxonin using the *GAN* replacement vector [64]. Firstly, a strong promoter to drive the expression of *GAN* may result in the loss of normal IF cytoskeleton, due to an overexpression of gigaxonin protein. As gigaxonin is expressed at very low levels in humans (~7500 molecules/lymphoblast cell), the authors have used a weak promoter JeT to drive the expression of gigaxonin [65]. From the in vitro and in vivo results, the amount of gigaxonin production driven by the JeT promoter sufficiently reduced the amount of IF and NF aggregates without disrupting the normal network of IF and NFs required for function. Besides, using this construct for the vector keeps the opportunity to increase the dose in human patients to maximize therapeutic benefits as the risk of causing toxicity due to over-expression of gigaxonin protein is not anticipated. 

Previously, Gray et al. have also demonstrated that successful *GAN* gene replacement prevents the formation of IF aggregates in three distinct cell lines (fibroblasts) derived from persons with GAN as well as *GAN*-null mice [66]. In addition to this study, Wichterle et al. have shown a reduction in IF aggregates after successful *GAN* gene replacement in induced pluripotent stem cells (iPSCs) derived from three different persons with GAN [61].

Due to the promising results obtained in vitro and in vivo, a Phase I clinical trial to assess the safety of the vector is ongoing (ClinicalTrials.gov Identifier: NCT02362438).

## 5. Phase I Clinical Trial of GAN

A phase I first-in-human, open-label, non-randomized, single-dose escalation clinical trial to test the safety of gene transfer vector scAAV9/JeT-GAN in persons diagnosed with GAN has been initiated [67]. 21 participants currently enrolled in the trial will receive the treatment intrathecally to target the brain and the spinal cord. 

The primary goal of this study is to access the safety of the vector. Assessing disease symptoms upon treatment, examining neuropathology, investigating the presence of inflammatory markers in the cerebrospinal fluid, and analyzing vector shedding are secondary interests of the trial.

## 6. Concerns about AAV9-Mediated Gene Replacement Therapy for GAN

GAN is caused due to the lack of gigaxonin, a ubiquitously expressed protein. The symptoms of this neurodegenerative disorder manifest as vision loss and optic atrophy as well [61]. In Gray et al.’s study, they observed an accumulation of Ifs (i.e., glial fibrillary acidic protein) in lens epithelial cells of 4-month-old GAN KO mice and human specimens obtained from autopsy [68]. Additionally, in a rat model of GAN, extreme degeneration of the retinal rod and cone photoreceptors were observed. The AAV9-mediated gene replacement therapy for GAN, which is under a phase I clinical trial, does not report on the efficacy of the treatment on photoreceptor cell deterioration. Visual impairment in a person with GAN has not been studied concerning photoreceptor cell loss, however, as vision loss and optic atrophy are symptoms of concern it is important to investigate the efficiency of gene transfer in the human eye [69].

Additionally, the efficiency of AAV9 transduction in Schwann cells is still under investigation [70]. In a study by Gray et al., a low amount of AAV9 was detected in peripheral nerve samples ensheathed by Schwann cells and surrounding endoneurium and perineurium cells. However, they suggested further experiments to ensure that the vector is delivered in Schwann cells particularly as the positive signal detected in their experiments could potentially mean AAV9 was only delivered to the axons or only the endoneurium and perineurium cells [71]. Investigating the efficiency of AAV9 delivery in Schwann cells is of interest as, the GAN KO mouse model has severe IF accumulation of IFs in the cytoplasm of Schwann cells, associated with both myelinated and unmyelinated fibers. Hence, targeting Schwann cells for the treatment of GAN must be prioritized. In preclinical studies for developing AAV9-mediated gene replacement therapy for GAN, GAN KO mice that received treatment showed an increased number of normal-appearing Schwann cells [64].

Another big concern with AAV9-mediated gene replacement therapy for GAN is neurotoxicity. A study by Wilson et al. has shown a neuronal loss in dorsal root ganglion (DRG) in non-human primates upon administration of AAVs [72]. Besides, signs of DRG toxicity were observed in a person with GAN who received gene therapy [73].

For people with GAN, there is currently no approved treatment beyond counseling and symptomatic treatments, such as aquatic and physical therapy, bracing, or painkillers [74]. However, gene therapy for giant axonal neuropathy has shown efficacy in preclinical trials, and a first-in-human phase I clinical trial of intrathecal gene transfer for GAN has been initiated at the National Institutes of Health, NIH (NCT02362438) [64]. However, the effects were unclear, where autopsies of some participants showed signs of dorsal root ganglion (DRG) damage, indicative of AAV neurotoxicity [73].

## 7. Conclusions

GAN is a rare pediatric disease that has no cure at the moment. However, gene therapy for giant axonal neuropathy has shown efficacy in preclinical trials, and a first-in-human phase I clinical trial of intrathecal gene transfer for GAN has been initiated at the National Institutes of Health, NIH (NCT02362438) [64]. The clinical trial, which is expected to be completed by 2035, will hopefully result in a safe and effective treatment for GAN [67]. 

Additionally, gene editing and antisense oligonucleotide (ASO) mediated treatment are viable options for treating rare genetic diseases, like GAN, which affect the brain and the spinal cord [75,76,77]. Gene editing tools including zinc finger nucleases (ZFNs), transcription activator-like effector nucleases (TALENs), and CRISPR/Cas systems have been thoroughly studied in pre-clinical trials for the treatment of rare diseases [78,79,80]. However, very few have reached clinical trials for treating rare diseases affecting the CNS [81]. 

Another therapeutic tool called ASOs holds great potential when it comes to treating neurological diseases [82]. Nusinersen (Spinraza^®^) is an FDA-approved ASO-mediated drug prescribed for Spinal Muscular Atrophy (SMA), a neuromuscular disorder [83,84]. In a phase III clinical trial, infants treated with nusinersen showed greatly improved motor-milestone response and had a 47% decrease in the risk of fatality and the need for permanent assisted ventilation [85,86]. ASO-mediated treatment for Huntington’s disease (HD), a neurodegenerative disorder, is also under investigation [87]. Upon intrathecal administration, ASOs can be detected in high levels across the CNS and beneficial levels can be maintained with repeated doses [82]. As ASOs have a hopeful future in treating rare neurodegenerative diseases, investigating N-of-1 ASO-mediated treatment for GAN is encouraged [76]. Around 6.6% of people with GAN have mutations at splice donor/acceptor sites [7]. As most ASO-mediated therapies ameliorate mutations in splice sites, this population can particularly benefit from this venture. 

With remarkable advances made in the molecular biology field, several strategies to fight rare diseases like GAN are evolving. Although there is no cure for GAN at the moment, with the ongoing clinical trial and several other available strategies to treat neurological maladies, we can be hopeful for the patients and their families. 

## Figures and Tables

**Figure 1 jpm-13-00091-f001:**
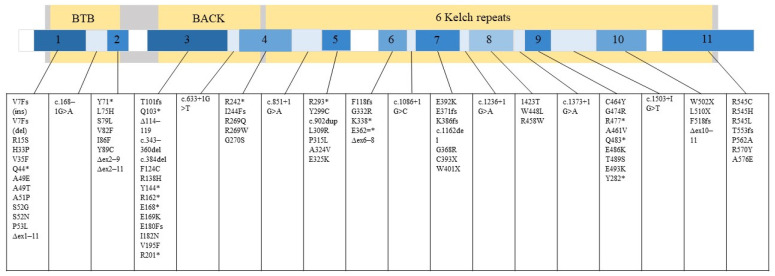
89 variants associated with GAN that can occur in either exon or intron are listed. Gigaxonin is illustrated by the large grey rectangle and its domains in yellow. The smaller rectangle depicts the *GAN* gene with its exons numbered 1–11. The relative number of mutations that can occur in different gene regions is represented in a blue gradient. Darker blue equals more mutation. White equals no mutation reported in that region.

**Table 1 jpm-13-00091-t001:** Summary of suggestive findings and clinical testing including peripheral nerve biopsy, electrophysiologic, and neuroimaging examinations available for diagnosis of giant axonal neuropathy.

Diagnostic Findings and Testing	Comments
Clinical findings	Early-onset peripheral motor and sensory neuropathy (all individuals) Infantile- to early childhood-onset CNS involvement include developmental delay/intellectual disability ○ Cerebellar, pyramidal, and cranial nerve signs with advancement in the disease Kinky hair that differs from parents (classical GAN)
Electron Microscopy	Abnormally large axons in their paranodal junction and decreased axonal myelination in sural nerve and skin/muscle fiber biopsy
Electrophysiology	NCS: Generally normal to moderately reduced nerve conduction velocity. Can prolong, even to demyelinated range EMG: Severely reduced compound motor action potentials and absent sensory nerve action potentials of distal limbs. Sensory evoked responses (EEG): Latency to deficit in auditory brainstem evoked, visual evoked, and somatosensory evoked responses Spikes and sharp waves (EEG): Epileptiform transient discharges in the form of focal spikes and sharp waves (regardless of presence or absence of a history of seizures)
Neuroimaging	MRI: **Classical GAN phenotype** Hyperintense signal in cerebellar white matter surrounding the dentate nucleus. High T2-weighted sequences (and decreased T1 signal) in anterior and posterior periventricular regions and cerebellar white matter. Increased T2-weighted signals also occurs in the internal capsule, thalamus, and brainstem **Milder phenotype** Normal (most cases) to mild cerebral and cerebellar atrophy MRS: Increased Cho compounds and osmolyte myoinositol (Ins). Decreased *N*-acetylaspartate (NAA)

NCS = nerve conduction study. EMG = electromyogram. EEG = electroencephalogram, MRI = magnetic resonance imaging. MRS = magnetic resonance spectroscopy.

**Table 2 jpm-13-00091-t002:** General categorization of clinical phenotypes observed in individuals with classical giant axonal neuropathy. Multiple systems are often involved to reflect a certain symptom observed.

System/Concern	Feature	
	Pyramidal signs (e.g., spasticity, Babinski signs)	
	Romberg sign	
	Cerebellar signs (e.g., ataxia, nystagmus, dysarthria)	
		Facial weakness
		Bulbar weakness
	Cranial nerve involvement	Ptosis
		Ophthalmoplegia
CNS		Optic atrophy
		Dysarthria
	Hearing loss	
	Vision loss	
	Vertigo	
	Epilepsy	
	Intellectual disability/developmental delay	
	Cognitive decline	
	Dementia	
		Decreased muscle strength/tone
	Motor neuropathy	Diffused muscle atrophy
		Distal motor weakness
PNS		Proximal motor weakness
	Sensory neuropathy	
	Flaccid paralysis	
	Areflexia	
	Constipation	
	Reflux	
ANS	Regurgitation	
	Diabetes	
	Renal tubular acidosis	
	Lactose intolerance	
	Kinky hair/ long eyelashes	
	Pale skin	
	High forehead	
Body-wide/miscellaneous	Short stature	
	Scoliosis (often associated with lumbar hyper-lordosis)	
	Tendon contracture	
	Foot deformity/arched feet	
	Precocious puberty	

## Data Availability

Not applicable.

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
