# Peer review of "Genetic Approaches for the Treatment of Giant Axonal Neuropathy"

_jpm, 2022, doi:10.3390/jpm13010091_

Round 1

Reviewer 1 Report

The authors introduced the rare disease (giant axonal neuropathy) caused by GAN mutations and discussed its potential therapeutic approaches. Overall, the manuscript is clear and clinically significant but requires more relevant genetic and pathological information as a review. I have the following comments.

1. Page 1 line 33: This sentence "According to the Leiden Variant Database, at least 150 different mutations associated with GAN have been identified." is not accurate. Some of these mutations are classified as benign or likely benign, which will not affect the function of GAN. In the HGMD database, there are only 79 variants associated with GAN

2. What is the clinical diagnosis for giant axonal neuropathy caused by GAN mutations? please add some pathological figures to show the differences between disease and healthy states.

3. Giant axonal neuropathy is an autosomal recessive disorder caused by loss of GAN function. The authors should introduce the GAN gene, its encoding protein, and its domains, and label the reported mutations in the gene. This work could help the audience understand the mutation spectrum and the hot-spot regions to better recognize the work of potential therapeutic approaches.

4. Page 2 Clinical heterogeneity in persons with GAN part: please add a table to summarize the phenotypic heterogeneity of reported GAN patients.

5.  Page 7 line 322 ASO part: currently, ASO therapy mostly targets aberrant splicing mutations, such as the SMA reported by the authors. For GAN disorder, the authors should first report the number of GAN mutations that are splicing mutations, then highlight that this subset of GAN mutations may be treatable with ASO.

Author Response

REVIEWER 1
1. Page 1 line 33: This sentence "According to the Leiden Variant Database, at least 150 different
mutations associated with GAN have been identified." is not accurate. Some of these mutations are
classified as benign or likely benign, which will not affect the function of GAN. In the HGMD
database, there are only 79 variants associated with GAN. 
Page 1 Line 33-36: According to the Leiden Open Variation Database (LOVD) (updated 2022), National
Center for Biotechnology Information (NCBI) (updated 2022), and the mutation summary reported by
Lescouzères, and Bomont, P (2020), there are 89 variants that lead to manifestation of GAN.
Thank you for pointing this out. We have complied the most recent information available on the
reported number of GAN mutations and referred to that in our paper.
2. What is the clinical diagnosis for giant axonal neuropathy caused by GAN mutations? please add
some pathological figures to show the differences between disease and healthy states.
Table 1 added. An individuals suspected with GAN can take molecular genetic testing for biallelic GAN
gene pathogenic mutation to confirm the diagnosis. However, there is not yet established clinical
diagnostic criteria for GAN-related neuropathy. Currently, suggestive findings and specialized testing
includes peripheral nerve biopsy, electrophysiologic, and neuroimaging examinations.
3. Giant axonal neuropathy is an autosomal recessive disorder caused by loss of GAN function. The
authors should introduce the GAN gene, its encoding protein, and its domains, and label the
reported mutations in the gene. This work could help the audience understand the mutation
spectrum and the hot-spot regions to better recognize the work of potential therapeutic
approaches.
Figure 1 has been added.
4. Page 2 Clinical heterogeneity in persons with GAN part: please add a table to summarize the
phenotypic heterogeneity of reported GAN patients.
Table 2 added.

5. Page 7 line 322 ASO part: currently, ASO therapy mostly targets aberrant splicing mutations, such as
the SMA reported by the authors. For GAN disorder, the authors should first report the number
of GAN mutations that are splicing mutations, then highlight that this subset of GAN mutations may
be treatable with ASO.
Page 9 Line 400-402: Page 9 Line 400-402 Around 6.6% of people with GAN have mutations at splice
donor/acceptor sites. As most ASO-mediated therapies ameliorate mutations in splice sites, this
population can particularly benefit from this venture.

Reviewer 2 Report

This well writen review leads on the genetic treatment of GAN. The paper is well organized and will be a good reference for postdoctoral students. In my opinion the paper is suitable for publication in  the present form.

Author Response

REVIEWER 2
This well written review leads on the genetic treatment of GAN. The paper is well organized and will be
a good reference for postdoctoral students. In my opinion the paper is suitable for publication in the
present form.
Thank you very much. We appreciate your positive feedback.